# Vaccination with CD47 deficient tumor cells elicits an antitumor immune response in mice

Yang Li[1,2,5], Mingyou Zhang[1,2,5], Xiaodan Wang[1,3], Wentao Liu[1,3], Hui Wang[2] & Yong-Guang Yang[1,2,3,4]*

Cancer cells are poorly immunogenic and have a wide range of mutations, which makes them unsuitable for use in vaccination treatment. Here, we show that elimination of CD47, a ligand for the myeloid cell inhibitory receptor SIRPα, from tumor cells by genetic deletion or antibody blocking, significantly improves the effectiveness of the immune response to tumour cells. In both solid and hematopoietic mouse tumor models, vaccination with tumor cells or tumor antigen-expressing cells, that lack CD47 or were pre-coated with anti-CD47 antibodies, achieved an antitumor immune response. The efficacy of this approach was synergistically enhanced when used in combination with anti-PD-1 antibodies. The induction of antitumor responses depends on SIRPα+CD11c+ DCs, which exhibit rapid expansion following introduction of CD47-deficient tumor cells. Our results indicate that CD47-deficient whole tumor cells can induce antitumor responses.

[1] Key Laboratory of Organ Regeneration and Transplantation of the Ministry of Education, The First Hospital, and Institute of Immunology, Jilin University, Changchun, China. [2] Columbia Center for Translational Immunology, Columbia University Medical Center, New York, NY, USA. [3] National-local Joint Engineering Laboratory of Animal Models for Human Diseases, Changchun, China. [4] International Center of Future Science, Jilin University, Changchun, China. [5] These authors contributed equally: Yang Li, Mingyou Zhang. *email: yy2324@columbia.edu

mmunotherapy aims to achieve or provoke an immune response that targets molecules specifically expressed on cancer cells; and cancer vaccines represent one of the promising treatment strategies[1]. Cancer vaccines induce tumor-specific immune responses and thus, may act synergistically with immune checkpoint blockade or other immune therapeutics to boost antitumor immunity without elevating autoimmune responses. Enormous effort has been invested in developing effective cancer vaccines by identifying tumor-specific antigens[2,3]. Although recent progress in next-generation sequencing and novel bioinformatics has significantly shortened the time needed for mapping tumor-specific antigens[4,5], current technologies are still not very effective in delivering personalized therapy in a sufficiently timely manner because cancer cells exhibit highly distinct compositions of mutations with limited overlaps between patients. Furthermore, it is also very challenging to design and produce cancer vaccines that effectively target multiple tumor antigens, which is considered more effective than targeting a single tumor antigen[2,3].

Whole tumor cell vaccines have the potential to induce broad immune responses to multiple tumor antigens, including those not yet identified. Upon examining the responses of a large pool of patients with different solid tumor types, the rate of objective clinical responses was found to be significantly higher in patients receiving whole tumor cell-based immunotherapies than in those enrolled in immunotherapy with molecularly defined antigens[6]. However, because most tumor cells are poorly immunogenic, the efficacy of whole tumor cell vaccination is still unsatisfactory[7,8]. For this reason, allogeneic tumor cells and tumor-dendritic cell (DC) hybrids have been increasingly used to improve tumor antigen presentation[9,10]. New strategies are urgently needed to enhance the immunogenic potential of tumor cell vaccination.

CD47 is an ubiquitously expressed molecule that serves as a ligand for the immune inhibitory receptor, signal regulatory protein (SIRP)α, which is expressed on myeloid cells, including macrophages and DCs. CD47 provides a do not eat me signal to macrophages through SIRPα to prevent phagocytosis[11], so that macrophages mediate robust rejection of CD47-deficient cells[11] or xenogeneic cells expressing recipient SIRPα-incompatible CD47[12,13]. CD47-SIRPα signaling also regulates DC endocytosis[14,15] and activation[16,17]. In a mouse model of hepatocyte transplantation, we recently observed that intrasplenic (i.s.) injection of CD47-deficient hepatocytes induces rapid and robust innate immune cell activation[18] and provokes T cell immune responses specific for a donor minor antigen[19]. Thus, we propose that injection of CD47-deficient tumor cells or cells carrying tumor antigens (TAs) may induce strong antitumor immunity. Here we show that in both mouse melanoma and lymphoma models, strong and specific antitumor immune responses were induced by injection of CD47-deficient, but not CD47-competent, tumor cells or TA-expressing normal cells. A significant antitumor protection was also detected in mice that were vaccinated with anti-CD47 antibody-coated tumor or TA-expressing cells. Furthermore, the vaccinated mice were protected from rechallenge with the same tumor cells, demonstrating the development of immunological memory and sustained antitumor responses. Mechanistic studies revealed that rapid expansion of SIRPα+ CD11c+ DCs following CD47KO tumor cell vaccination is essential for induction of antitumor responses in these mice. Our study highlights that CD47-deficient whole tumor cell vaccine is effective in eliciting antitumor immune responses.

## Results

### Antitumor effect of CD47KO melanoma cell vaccine.
CD47KO B16F0 melanoma cells were generated using the CRISPR-Cas9 technique (Fig. 1a). In order to prevent tumor cell proliferation, CD47KO or WT B16F0 cells were treated with 70 Gy irradiation or mitomycin C immediately before injection into the recipient mice. Immunofluorescence microscopy revealed that the injected tumor cells, regardless of CD47 deficiency, were clearly detected in the spleen (localized mainly at the injection site) and liver for at least 24 h following i.s. injection (Supplementary Fig. 1), consistent with previous studies where intrasplenic injection was used to establish orthotopic hepatocellular carcinoma and hepatic metastases in mice[20,21].

All mice were challenged by subcutaneous injection of B16F0 melanoma cells 7 days after i.s. tumor cell vaccination. Melanoma tumor growth was significantly delayed in mice immunized with CD47KO B16F0 cells compared to those immunized with control B16F0 cells, and the latter showed no significant protection compared to the PBS controls (Fig. 1b, c). Furthermore, T cells from mice vaccinated with CD47KO B16F0 cells showed an increased IFN-γ production (Fig. 1d) and enhanced killing of tumor target cells (Supplementary Fig. 2a) compared to those from mice receiving medium or WT B16F0 cell vaccination. Accordingly, an increase in CD44hiCD62Llow/− effector T cells was detected in both CD4+ and CD8+ T cell populations in mice vaccinated with CD47KO tumor cells compared to those immunized with PBS or WT tumor cells (Supplementary Fig. 2b).

To mimic a clinical setting more closely, we next evaluated the efficacy of tumor cell vaccination in mice that had been pre-inoculated with tumor cells. Again, vaccination with irradiated CD47KO tumor cells significantly suppressed tumor growth in mice pre-inoculated with B16F0 compared to those injected with media (Fig. 1e). Furthermore, significantly suppressed tumor growth was detected when tumor-free mice from the CD47KO B16F0-vaccinated group were re-challenged with B16F0 tumor cells, demonstrating the development of memory anti-tumor responses in these mice (Fig. 1f). Because CRISPR-Cas9 gene editing is highly promising for clinical therapies[22,23], the use of CD47KO tumor cells for vaccination is considered a clinically practical treatment.

### Antitumor responses induced by TA-expressing CD47KO cells.
We next sought to induce antitumor responses with intrasplenic injection of normal cells expressing tumor antigens. We compared antitumor responses against OVA-expressing lymphoma EG7 in mice following i.s. injection of splenocytes from OVA-transgenic (Tg) wild-type (WT) or CD47KO mice. WT C57BL/6 (B6) mice received i.s. injection of PBS, or WT or CD47KO OVA-Tg B6 mouse splenocytes, and were then challenged with OVA+ lymphoma EG7 cells (i.v.; $2.5 \times 10^6$ per mouse). Compared to PBS-injected controls that all died of tumor by 47 days, injection of CD47KO, but not WT OVA-Tg mouse splenocytes achieved potent antitumor responses (Fig. 2a). Mice receiving WT splenocytes showed no significant improvement in survival compared to PBS-injected controls, while 60% of the mice receiving CD47KO splenocytes survived long-term. With reduced magnitude, mice vaccinated by intravenous (i.v.) injection of CD47KO OVA-Tg splenocytes also showed significantly improved antitumor responses against EG7 and prolonged survival (Fig. 2b). Vaccination with CD47KO OVA-Tg mouse splenocytes significantly improved the survival of mice that had been pre-inoculated with EG7 tumor cells, with 40% of the mice surviving through the observation period of 100 days, compared to the controls that all died within 70 days (Fig. 2c).

To determine whether CD47KO tumor vaccine-induced antitumor immunity is antigen-specific and able to mediate durable protection, we re-challenged the long-term survivors with EG7 or OVA-negative EL4 tumor cells. Compared with the

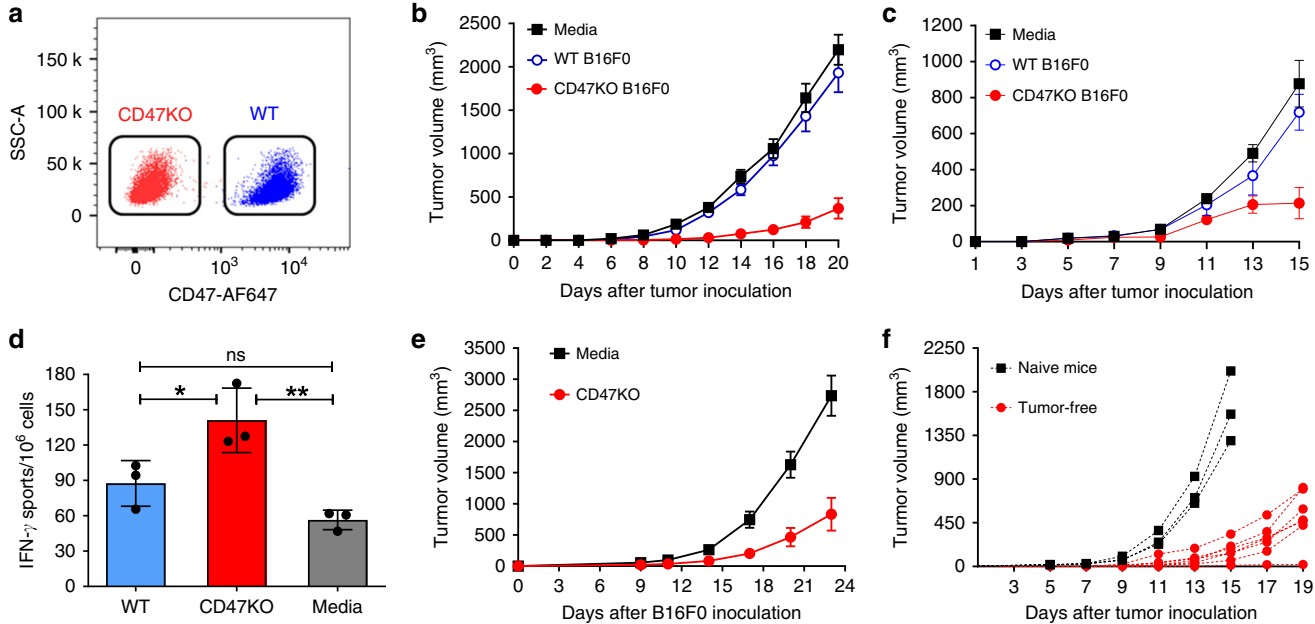

**Fig. 1 Antitumor responses achieved by CD47KO tumor cell vaccination. a** Representative flow cytometry profiles showing CD47 expression on wild-type (WT) and CD47KO B16F0 cells. CD47KO cells were generated by CRISPR/Cas9-targeted inactivation of cd47 gene. **b** B6 mice were injected (i.s.) with media ($n = 19$), or with irradiated WT ($n = 16$) or CD47KO ($n = 20$) B16F0 cells, followed 7 days later by subcutaneous injection of $2 \times 10^5$ WT B16F0 melanoma cells. **c** B6 mice were injected (i.s.) with media ($n = 4$ at days 11 and 13; $n = 9$ at other time points), or with WT ($n = 4$ at days 11 and 13; $n = 7$ at all other time points) or CD47KO ($n = 5$ at days 11 and 13; $n = 10$ at all time points) B16F0 cells that were treated with mitomycin C (at 40 µg/mL for 2h), followed 7 days later by subcutaneous injection of $2 \times 10^5$ WT B16F0 melanoma cells. **d** Spleen cells were harvested from mice 7 days after vaccination with irradiated WT or CD47KO B16F0 cells or media, and IFN-γ production was measured by ELISPOT assay ($n = 3$ per group). Results are presented as mean ± SD (*$p < 0.05$, **$p < 0.01$; unpaired $t$-test). **e** B6 mice were pre-inoculated subcutaneously with $1 \times 10^5$ WT B16F0 melanoma cells, followed 1 day later by vaccination (i.s.) with media ($n = 6$) or irradiated CD47KO B16F0 cells ($n = 5$). **f** Tumor-free mice in the group vaccinated with irradiated CD47KO B16F0 cells 30 days after first tumor cell injection were re-challenged with $2 \times 10^5$ B16F0 cells ($n = 7$). Age-matched naïve mice were used as the controls ($n = 3$). Each symbol represented an individual mouse. Source data are provided as a Source Data file.

control group (i.e., age-matched naïve B6 mice that had not received tumor cell injection) that all died of tumor within 45 days after EG7 tumor cell injection, significantly improved survival with 70% durable protection was seen in CD47KO OVA-Tg mouse splenocyte-injected mice that survived the initial EG7 challenge (Fig. 2d). The antitumor immunity was antigen-specific as CD47KO splenocyte-injected mice showed no improvement in antitumor responses against EL4 (OVA-) tumor cells compared to the controls (Fig. 2e). This was further supported by an in vivo killing assay, in which OVA+, but not OVA−, cells were significantly killed in mice receiving CD47KO OVA-Tg mouse splenocytes (Supplementary Fig. 3). Together, these results demonstrate that i.s. vaccination with CD47KO tumor antigen-expressing splenocytes achieves potent and durable antigen-specific antitumor immunity.

**Effect of anti-CD47 Ab-coated TA⁺ cell or tumor cell vaccine.** We next assessed the antitumor immunity in mice following injection of WT splenocytes that were pre-coated with anti-CD47 antibody (Ab). Briefly, WT OVA-Tg mouse splenocytes were incubated with anti-CD47 mAb at a pre-titrated concentration (20 µg/mL; Fig. 3a) for 1 h, and then intrasplenically injected into WT B6 mice. Mice receiving an i.s. injection of PBS or CD47KO OVA-Tg mouse splenocytes were used as negative and positive controls, respectively, and antitumor responses were measured by EG7 challenge 7 days later. Although the magnitude seemed to be lower than that achieved with CD47KO OVA-Tg mouse splenocytes, injection of anti-CD47 Ab-coated WT splenocytes also achieved significant antitumor responses compared to the PBS-injected controls (Fig. 3b).

We also assessed the potential to enhance antitumor effects of vaccination with anti-CD47 Ab-coated irradiated tumor cells in mice challenged with two different sublines of murine melanoma, B16F0 and B16F10. Like the observation in the EG7 model, B16F0 melanoma growth was significantly delayed in mice vaccinated (i.s.) with anti-CD47 Ab-coated irradiated B16F0 cells compared to those vaccinated with irradiated B16F0 cells or medium, although to a lesser extent than CD47KO tumor-vaccinated mice (Fig. 3c). Similar results were observed in the B16F10 model, in which vaccination (i.s.) with anti-CD47 Ab-coated, but not non-coated, tumor cells, induced a significant antitumor response against B16F10 melanoma compared to the medium controls (Fig. 3d). The lower protection by anti-CD47-coated WT tumor cells than CD47KO tumor cells may possibly be due to quick antibody dissociation from CD47 on tumor cells after intrasplenic injection into mice. In support of this possibility, an improved antitumor response was detected in mice receiving subcutaneous vaccination with anti-CD47-coated B16F10 tumor cells (Fig. 3e). These results also demonstrate that subcutaneous vaccination with tumor cells lacking surface CD47 can also induce significant antitumor responses. Thus, vaccination with antibody-coated tumor cells represents a novel approach to cancer immunotherapy.

**DC activation following CD47KO tumor cell vaccination.** Considering the potential of CD47-SIRPα signaling in regulation of DC function[16,17], we analyzed the changes in SIRPα⁺ DCs following tumor cell vaccination. Flow cytometry analysis at 24 h after tumor cell vaccination revealed that SIRPα⁺ cells consisted of mainly CD11c⁻CD11b⁺ non-DC myeloid cell and CD11c⁺

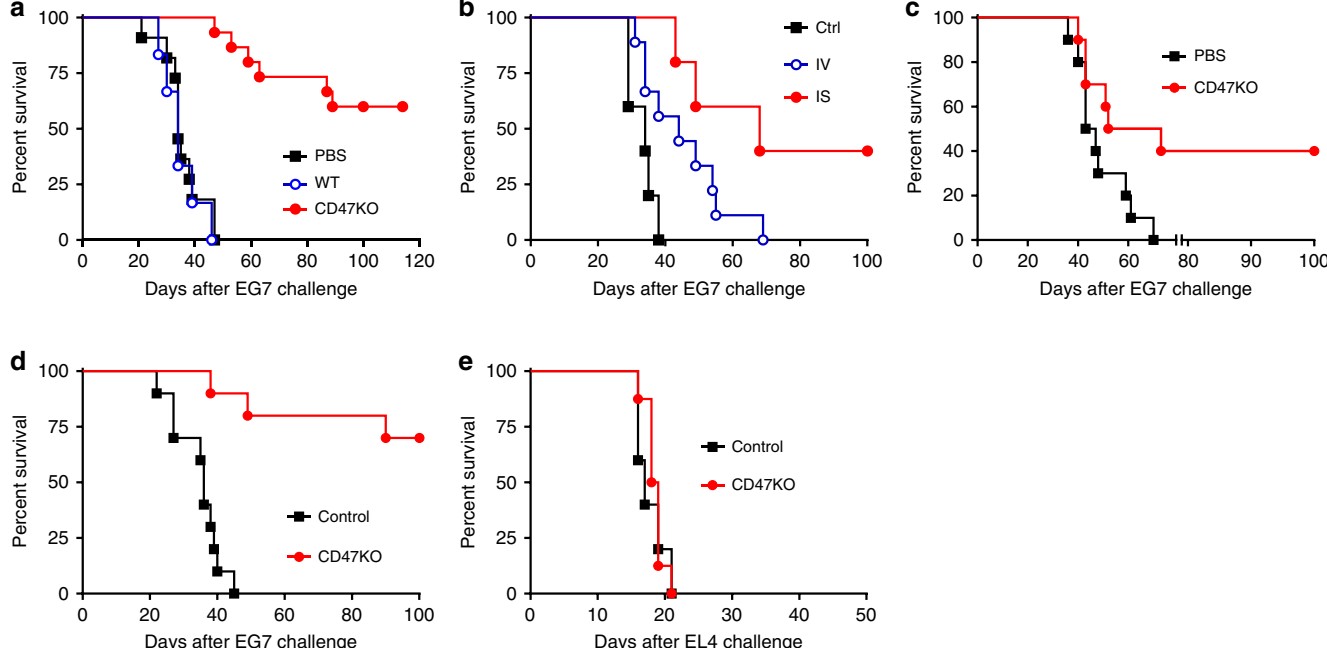

**Fig. 2 Injection of OVA-expressing CD47KO cells induces OVA-specific antitumor responses. a** C57BL/6 mice received intrasplenic injection of PBS ($n = 11$) or splenocytes from WT ($n = 6$) or CD47KO ($n = 15$) OVA-Tg C57BL/6 mice, followed 7 days later by injection (i.v.; $2.5 \times 10^6$) of EG7 cells. **b** C57BL/6 mice received intravenous (i.v.; $n = 9$) or intrasplenic (i.s.; $n = 6$) injection of CD47KO OVA-Tg splenocytes, or intrasplenic injection of PBS (Ctrl; $n = 5$), followed 7 days later by injection (i.v.) of $2.5 \times 10^6$ EG7 cells. **c** C57BL/6 mice were pre-inoculated intravenously with $1 \times 10^5$ EG7 cells, followed 1 day later by injection (i.s.) with PBS ($n = 10$) or CD47KO OVA-Tg splenocytes ($n = 10$). **d, e** CD47KO OVA-Tg splenocyte-vaccinated B6 mice that survived first EG7 challenge (CD47KO) were re-challenged (i.v.) with $2.5 \times 10^6$ EG7 ($n = 10$; **d**) or EL4 ($n = 8$; **e**) tumor cells. Controls are age-matched B6 mice challenged with EG7 ($n = 10$; **d**) or EL4 ($n = 5$; **e**) tumor cells. Source data are provided as a Source Data file.

DC populations, and the latter can be further divided into CD11b$^+$ and CD11b$^-$ DC subsets (Fig. 4a). The levels of SIRPα expression on these cells were significantly different with highest expression on CD11c$^+$CD11b$^+$ DCs, followed by CD11c$^+$CD11b$^-$ DCs and lowest expression on CD11c$^-$CD11b$^+$ myeloid cells (Figs. 4a, b).

Although SIRPα levels remained unchanged (Fig. 4a, b), a rapid expansion in SIRPα$^+$ cells was detected in mice vaccinated with CD47KO B16F0 cells compared to those receiving i.s. injection of WT B16F0 cells (Fig. 4c). It appeared that cell expansion was correlated with the levels of SIRPα expression. The ratios of SIRPα$^+$CD11c$^-$CD11b$^+$ non-DC myeloid cells that express the lowest levels of SIRPα were comparable between CD47KO and WT B16F0-vaccinated mice (Fig. 4d), but the former group showed a significant increase in SIRPα$^+$CD11c$^+$ DCs, including both CD11b$^+$ and CD11b$^-$ DC subsets (Fig. 4e). Similarly, a significant expansion in CD11c$^+$ DCs was also observed in mice following i.s. injection of CD47KO splenocytes (Supplementary Fig. 4). However, unlike vaccination with melanoma cells (Fig. 4), most of the CD11c$^+$ cells were CD11b$^+$ in mice receiving i.s. injection of splenocytes (Supplementary Fig. 4).

Similar results were observed in an independently repeated experiment, in which macrophages were distinguished from DCs by F4/80 and CD64 expression[24] for better comparison of expansion and activation of SIRPα$^+$CD11c$^+$ DCs in mice vaccinated with WT vs. CD47KO tumor cells (Fig. 5a). Again, CD47KO tumor cell vaccination resulted in a significantly increased frequency of CD11c$^+$SIRPα$^+$ (F4/80$^-$CD64$^-$) DCs (Fig. 5b) and their expression of MHC class II I-Ab (Fig. 5c) compared to vaccination with WT tumor cells. However, there was no significant difference in the percentage of F4/80$^+$CD64$^+$

monocytes/macrophages in spleens of WT vs. CD47KO tumor cell-vaccinated mice (Fig. 5d). Similar results were obtained from a repeat experiment, in which mice vaccinated with CD47KO tumor cells showed a significant increase in the frequencies of total CD11c$^+$SIRPα$^+$ DCs (Fig. 5e), and activated I-Ab$^{hi}$SIRPα$^+$CD11c$^+$ (Fig. 5f), CD40$^+$SIRPα$^+$CD11c$^+$ (Fig. 5g), CD86$^+$SIRPα$^+$CD11c$^+$ (Fig. 5h) DCs in spleens compared to mice receiving WT tumor vaccination. These results confirmed that vaccination with CD47KO melanoma cells is more effective than WT melanoma cells in induction of SIRPα$^+$ DC expansion and activation.

Splenic DCs consist of mainly two conventional DC subsets, SIRPα$^+$ (XCR1$^-$) and XCR1$^+$ (SIRPα$^-$) DCs[24]. Thus, we further compared the ability of flow cytometry-sorted SIRPα$^+$ (XCR1$^-$CD11c$^+$I-Ab$^+$F4/80$^-$CD64$^-$) and XCR1$^+$ (SIRPα$^-$CD11c$^+$I-Ab$^+$F4/80$^-$CD64$^-$) DCs (Supplementary Fig. 5) to stimulate T cell activation. Briefly, CD3$^+$CD8$^+$ T cells isolated from OT-I mice were labeled with CellTrace™ Violet and cultured with 1nM OVA peptide-pulsed DCs for 3 days, then T cell responses were assessed by measuring T cell proliferation (i.e., CellTrace™ Violet fluorescence dilution) and CD25 expression (Fig. 6a). There was a significant increase in the percentage of Violet$^{-/low}$ cells (Fig. 6b) and decrease in Violet fluorescence intensity (Fig. 6c) in CD8 T cells stimulated by SIRPα$^+$ DCs compared to those stimulated by XCR1$^+$ DCs. Furthermore, a significantly increased frequency of CD25$^+$ cells (Fig. 6d), with more pronounced CD25 upregulation (Fig. 6e) and Violet fluorescence dilution (Fig. 6f) was observed in Violet$^{-/low}$ CD8 T cells stimulated by SIRPα$^+$ DCs compared to those stimulated with XCR1$^+$ DCs. Similar results were observed in a repeated experiment, in which DCs were pulsed with OVA peptides at a lower concentration (100 pM; Supplementary Fig. 6). These results indicate that SIRPα$^+$ DCs were significantly more effective than SIRPα$^-$ DCs in

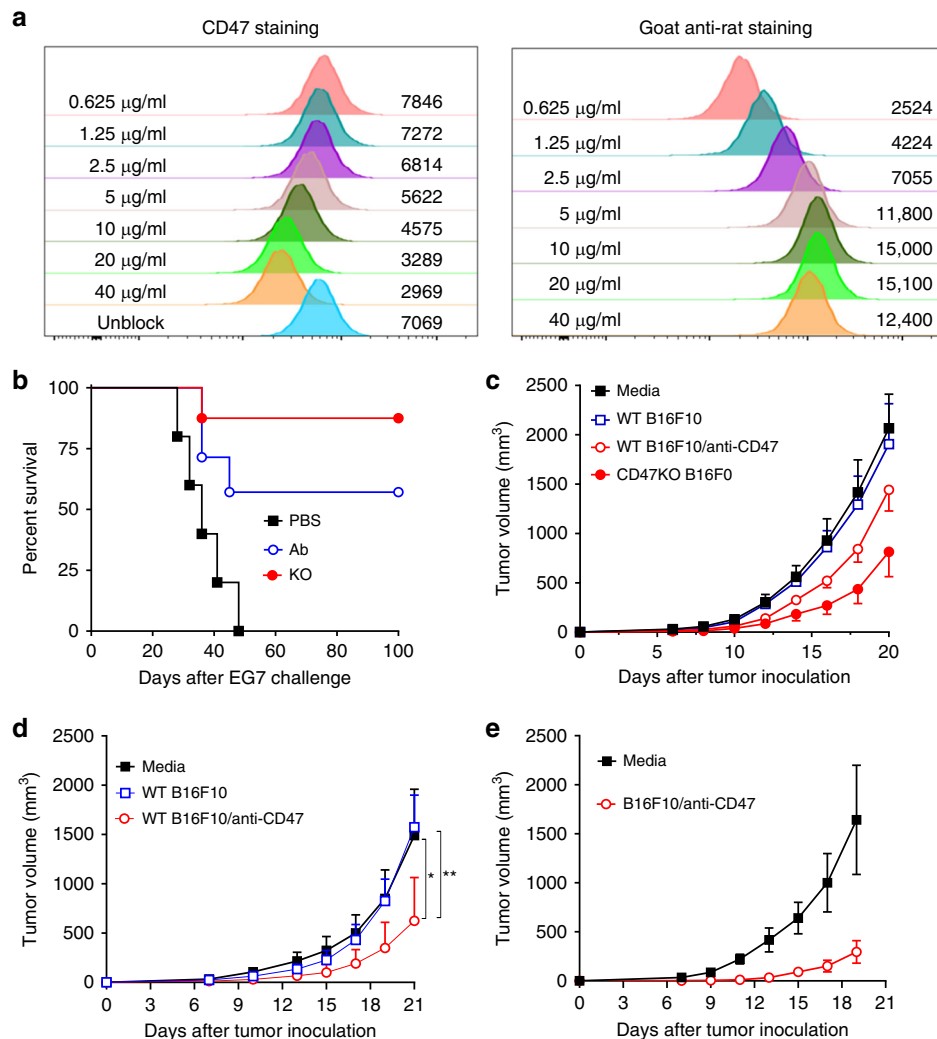

**Fig. 3 Antitumor effects of anti-CD47 Ab-coated WT tumor or tumor TA-expressing cells. a** Titration of anti-CD47 antibody for blocking CD47 on tumor cells. B16F0 WT cells were incubated with 2-fold serially diluted rat anti-mouse CD47 mAb MIAP301 (blocking; rat IgG) at 4 °C for 1 h, then stained with AF647-conjugated anti-CD47 mAb MIAP301 (left) or with AF647-conjugated goat anti-rat IgG (right). The numbers in the left and right sides of each histogram denote the concentration of anti-CD47 blocking mAb and MFI of secondary staining with AF647-conjugated antibodies. **b** Survival of mice that received intrasplenic injection of PBS ($n = 5$), 20 μg/mL anti-CD47 Ab-coated WT OVA-Tg splenocytes ($n = 7$) or CD47KO OVA-Tg splenocytes ($n = 8$), followed 7 days later by injection (i.v.; $2.5 \times 10^6$) of EG7 cells. **c** Tumor growth in mice that were intrasplenically vaccinated with media ($n = 5$), or with irradiated WT B16F0 ($n = 5$), WT B16F0 coated with anti-CD47 antibody (20 μg/mL; $n = 8$), or CD47KO B16F0 ($n = 7$) cells, followed 7 days later by subcutaneous injection of $2 \times 10^5$ WT B16F0 melanoma cells. **d** Tumor growth in mice that were intrasplenically vaccinated with media ($n = 8$), or with irradiated WT B16F10 ($n = 7$) or anti-CD47 Ab-coated WT B16F10 (20 μg/mL; $n = 7$), followed 7 days later by subcutaneous injection of $1 \times 10^5$ WT B16F10 melanoma cells. **e** Tumor growth in mice that were subcutaneously vaccinated with media ($n = 5$) or with irradiated anti-CD47 Ab-coated WT B16F10 (20 μg/mL; $n = 10$), followed 7 days later by subcutaneous injection of $2 \times 10^5$ WT B16F10 melanoma cells. Tumor growth is shown as tumor volumes (mean ± SEM) over time. Source data are provided as a Source Data file.

inducing CD8 T cell proliferation and activation, providing a mechanistic link between expansion/activation of SIRPα+CD11c+ DCs and enhanced antitumor T cell responses in mice vaccinated with CD47-deficient tumor cells.

**Effect of CD47KO tumor cell vaccine depends on CD11c+ DCs.** We then assessed antitumor responses after CD47KO tumor vaccination in mice with or without CD11c+ DC depletion using CD11c-DTR-Tg mice. To avoid DT-induced morbidity and mortality, we used CD11c-DTR bone marrow (BM) chimeras, in which DTR is expressed only on CD11c+ hematopoietic cells (Supplementary Fig. 7)[16]. DC depletion in CD11c-DTR BM chimeras was achieved by injection of DT at 12 h prior to and 24 h after tumor cell vaccination (Fig. 7a). Flow cytometry

analysis of splenocytes from randomly selected mice confirmed depletion of CD11c+ cells by DT treatment (Fig. 7b). Vaccination with CD47KO B16F0 melanoma cells significantly delayed tumor growth in the CD11c+ cell replete, but not in CD11c+ DC-depleted chimeras compared to the controls receiving no tumor cell vaccination (Fig. 7c). CD11c+ DC depletion in DT-treated CD11c-DTR B6 mice and CD11c-DTR BM chimeras lasts for 2 days[16,25,26]. Thus, the abrogation of the antitumor responses in DT-treated mice indicates that the early activation of SIRPα+CD11+ DCs plays an important role in priming antitumor T cells following CD47KO tumor vaccination.

**Synergy between CD47-deficient tumor vaccine and PD-1 Ab.** We next attempted to enhance the antitumor effect of

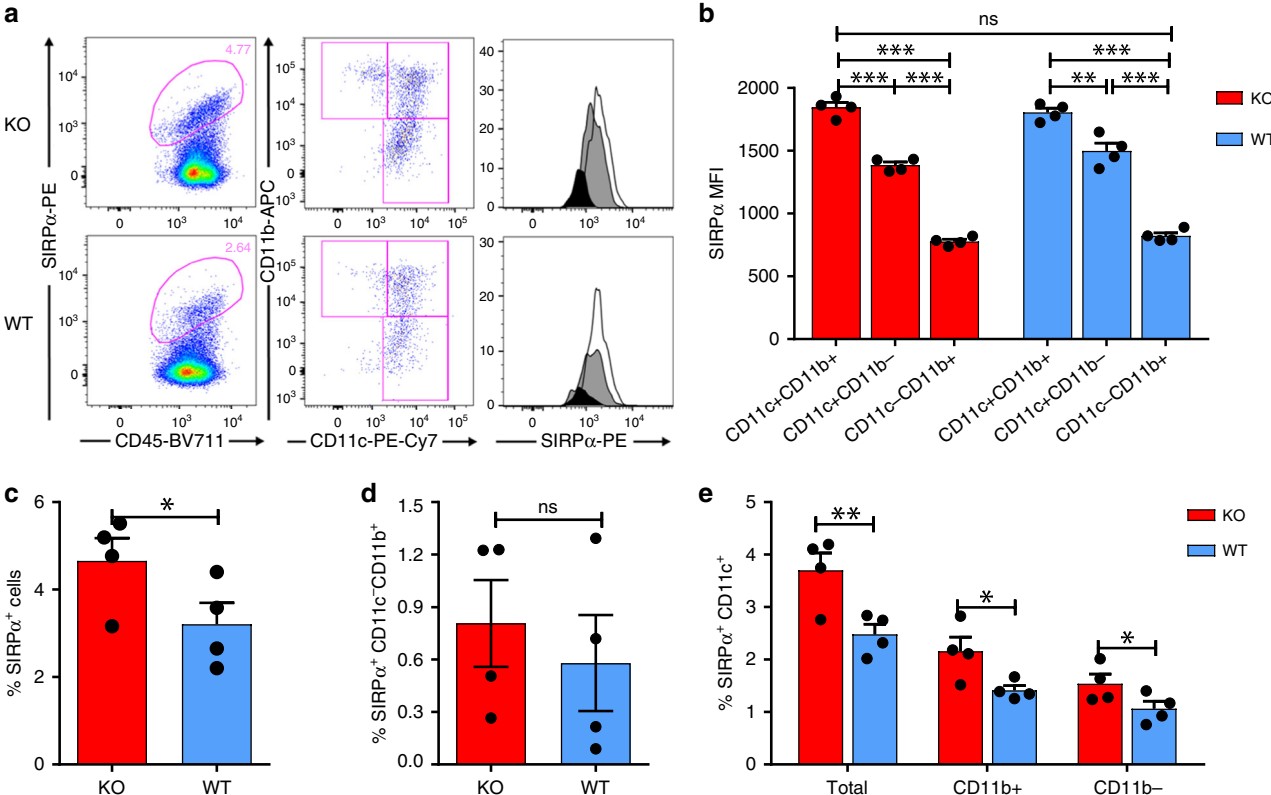

**Fig. 4 Phenotypes of SIRPα⁺ cells and levels of SIRPα expression.** Spleen cells harvested 24 h after vaccination with $2.5 \times 10^6$ irradiated WT or CD47KO B16F0 cells were analyzed for SIRPα expression on various cell populations by flow cytometry. **a** Representative flow cytometry profiles showing the ratio of SIRPα⁺ cells (left), CD11b and CD11c expression gated SIRPα+ cells (middle), and levels of SIRPα expression on gated CD11b⁺CD11c⁺ DCs (open histogram), CD11b⁻CD11c⁺ DCs (gray shaded histogram) and CD11b⁺CD11c⁻ myeloid cells (black shaded histogram; right). **b** MFI of SIRPα expression on the indicated cell populations. **p < 0.01; ***p < 0.005 (one-way ANOVA). **c** Percentages of total SIRPα⁺ cells. **d** Percentages of SIRPα⁺CD11b⁺CD11c⁻ myeloid cells. **e** Percentages of total SIRPα⁺CD11c⁺, SIRPα⁺CD11c⁺CD11b⁺, and SIRPα⁺CD11c⁺CD11b⁻ DC subsets. Results are presented as mean ± SEM ($n = 4$ per group). *p < 0.05; **p < 0.01; ***p < 0.005; ns, not significant (unpaired t-test). Source data are provided as a Source Data file.

CD47-deficient tumor cell vaccination by anti-PD-1 antibody in a therapeutic setting, in which tumor was established by subcutaneous injection of B16F10 cells ($5 \times 10^4$ per mouse) 3 days before tumor cell vaccination, and some groups of mice were treated with 4 injections (i.p.) of anti-PD-1 antibody (clone RMP1-14; 250 μg per injection at days 7, 10, 13, and 16; Fig. 8a). Although the average of tumor volumes was reduced in mice receiving vaccination with irradiated B16F10 tumor cells plus anti-PD-1 antibody compared those treated with anti-PD-1 antibody alone, the difference did not reach statistical significance at any time point (Fig. 8b). However, tumor growth was significantly delayed in mice vaccinated with anti-CD47 Ab-coated B16F10 cells (without anti-PD-1 antibody) compared to anti-PD-1 antibody-treated mice (Fig. 8b). Importantly, the antitumor effect was further significantly enhanced when anti-CD47 Ab-coated tumor cell vaccination was given in combination with anti-PD-1 antibody (Fig. 8b), demonstrating a synergistic effect between the two therapies.

## Discussion

Therapeutic cancer vaccines have shown limited efficacy in clinical trials. The development of effective cancer vaccines is hampered by multiple factors including the poor immunogenicity of the tumor cells and the high heterogeneity of tumor-associated antigens in human cancers, plus the deleterious tumor-associated immunosuppressive environment[2,3,7]. With the use of mouselymphoma and melanoma models, we demonstrated that elimination of CD47

expression from tumor antigen-expressing normal cells or tumor cells by genetic deletion or antibody blocking significantly improved the effectiveness of whole tumor cell vaccination. Furthermore, the efficacy can be further synergistically enhanced by anti-PD-1 antibodies. Because whole tumor cell vaccination may elicit broad immune responses against multiple tumor antigens in a personalized manner, this approach is possibly applicable to all immunogenic tumors regardless of whether the tumor antigens are identified.

CD47 is a ligand for signal regulatory protein (SIRP)α that is expressed by macrophages and DCs[11,14,15]. CD47 is known as a marker of self for macrophages, and its expression is required for normal cells to inhibit phagocytosis[11,14]. Elevated CD47 expression has been reported in various cancer cells, in which CD47 upregulation provides an important mechanism to evade macrophage killing, and is hence an adverse prognostic factor[27–29]. Accordingly, treatment (i.v.) with anti-CD47 antibodies was effective in various tumor models, with the efficacy largely macrophage-dependent[27,28,30–33]. However, CD47 is also expressed on normal cells, raising concern for the phagocytosis of red blood cells and platelets following i.v. injection of anti-CD47 antibodies[33,34]. Intratumoral injection was recently attempted in a mouse study, in which administration of anti-CD47 antibody induces antitumor responses without detectable adverse toxicity on normal cells[35]. In this model, the antitumor response was associated with STING expression in CD11c⁺ cells and was largely T cell-dependent[35]. In line with that study, here we found that vaccination with CD47KO tumor cells or TA-expressing

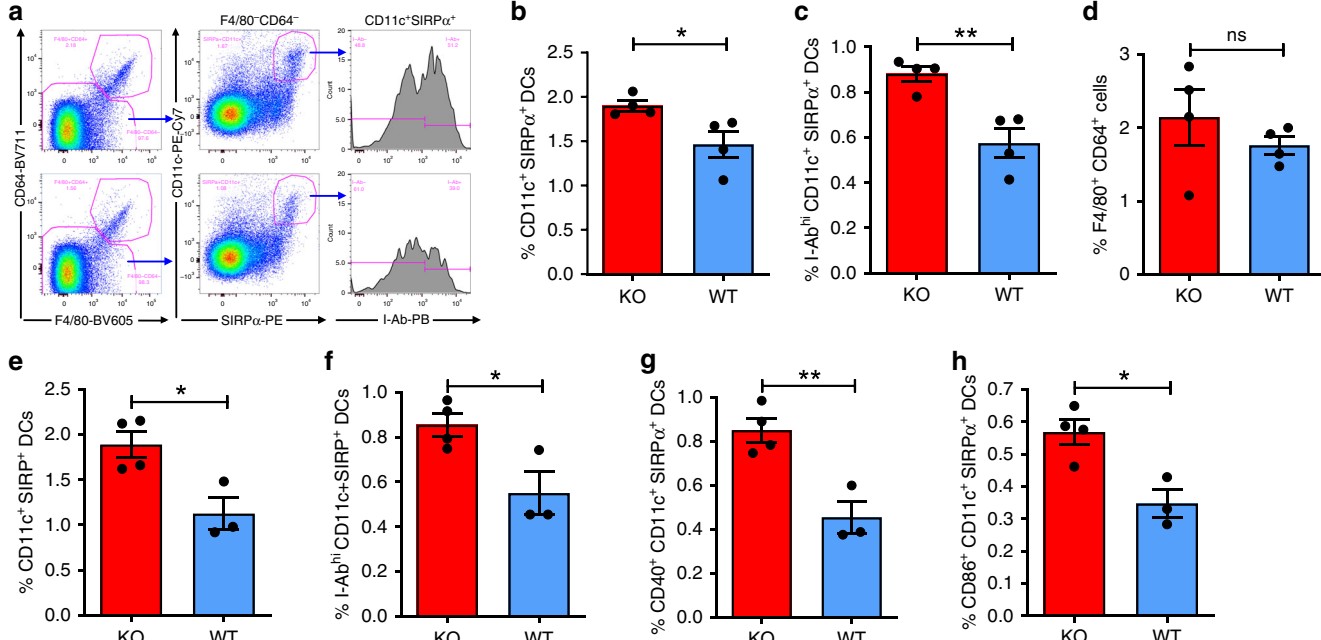

**Fig. 5 CD47KO tumor vaccination induces CD11c⁺SIRPα⁺ DC activation.** Spleen cells were harvested 24 h (**a–d**) or 60 h (**e–h**) after vaccination with 2.5 × 10⁶ irradiated WT or CD47KO B16F0 cells and analyzed for SIRPα⁺ DCs by flow cytometry. **a** Representative flow cytometry profiles showing expression of F4/80 and CD64 in spleen cells (left), SIRPα and CD11c expression in gated F4/80⁻CD64⁻ cell population (middle), and I-Ab expression in gated SIRPα⁺CD11c⁺ (F4/80⁻CD64⁻) DCs (right). Cells from CD47KO and WT tumor cell-vaccinated mice are shown in the top and bottom, respectively. **b–d** Percentages of SIRPα⁺CD11c⁺ (F4/80⁻CD64⁻) DCs (**b**), I-AbhiSIRPα⁺CD11c⁺ (F4/80⁻CD64⁻) DCs (**c**), and F4/80⁺CD64⁺ monocytes/ macrophages (**d**) among total spleen cells from CD47KO or WT tumor cell-vaccinated mice ($n = 4$ per group). **e–h** Percentages of total CD11c⁺SIRPα⁺ (F4/80⁻) (**e**), I-AbhiSIRPα⁺CD11c⁺ (**f**), CD40⁺SIRPα⁺CD11c⁺ (**g**), and CD86⁺SIRPα⁺CD11c⁺ (**h**) DCs in spleen cells from CD47KO ($n = 4$) or WT ($n = 3$) tumor cell-vaccinated mice (SIRPα⁺ DCs are defined as SIRPα⁺CD11c⁺F4/80⁻). Results are presented as mean ± SEM (*$p < 0.05$; **$p < 0.01$; ns, not significant; unpaired *t*-test). Source data are provided as a Source Data file.

splenocytes induced a rapid CD11c⁺ DC activation, and vigorous T cell-mediated antitumor responses. Our approach not only avoids the adverse effects of CD47 blockades, but also applies to patients for whom intratumoral injection is impossible.

SIRPα⁺ DCs are present in both lymphoid and non-lymphoid tissues[36]. We have previously shown that injection of allogeneic CD47KO cells induces rapid activation of SIRPα⁺ DCs with the majority expressing a CD11c⁺CD8⁻ phenotype in WT mice, which was associated with a stimulation of anti-donor T cell responses[16,17]. However, there has been limited information about the function of SIRPα⁺ DCs and their roles in cancer immunotherapy. A recent study showed that CD103⁺ DCs contribute to antitumor responses induced by PD-L1 blockades[37]. In line with that report, it was found that CD103⁺ DCs, in particular CD103⁺SIRPα⁺ DCs exhibit strong endocytic capacity[38].

Here we show that among the two major splenic conventional DC subsets, SIRPα⁺XCR1⁻ and SIRPα⁻XCR1⁺ DCs, the former was significantly superior to the latter in inducing antigen-specific T cell responses. SIRPα⁺ cells in the spleen are comprised of CD11c⁺CD11b⁺ and CD11c⁺CD11b⁻ DC subsets, and a CD11c⁻CD11b⁺ non-DC myeloid cell population. The SIRPα⁺ DCs express a significantly higher level of SIRPα than the SIRPα⁺ non-DC cells. Within the SIRPα⁺ DC population, SIRPα expression was significantly higher in CD11c⁺CD11b⁺ than CD11c⁺CD11b⁻ cells. Interestingly, vaccination with CD47-deficient tumor cells induced a significantly increased expansion and activation of CD11c⁺ SIRPα⁺ (F4/80⁻CD64⁻) DCs, but not of F4/80⁺CD64⁺ monocytes/macrophages. Increased endocytosis of CD47KO melanoma cells by CD11c⁺SIRPα⁺ DCs (Supplementary Fig. 8)

is assumed to be a mechanism for enhanced activation of these DCs following vaccination with CD47KO tumor cells. Using CD11c-DTR BM chimeras, we found that vaccination with CD47KO B16F0 cells could not stimulate antitumor immune responses in CD11c⁺ DC-depleted mice. Taken together, these results indicate that CD11c⁺SIRPα⁺ DCs play an essential role in the antitumor responses induced by CD47-deficient tumor cell vaccination. The finding also confirms that SIRPα is an important inhibitory receptor for SIRPα⁺ DCs, providing a molecular target that can be explored to modulate antigen presentation and T cell priming.

In summary, our results demonstrate that vaccination with CD47-deficient tumor cells induces strong antitumor immunity, revealing that removing cell surface expression of CD47 may significantly increase the immunogenicity of whole tumor cell vaccines. This study provides a proof-of-concept for achieving antitumor responses in a personalized manner by vaccination using cancer cells with antibody blocking or targeted deletion of CD47.

## Methods

**Animals.** Six- to eight-week-old female wild-type (WT), CD47 knock out (KO) C57BL/6 (B6), C57BL/6-Tg (CAG-OVA) 916Jen/J (OVA-Tg), and C57BL/6-Tg (TcraTcrb)1100Mjb/J (OT-I) mice were purchased from the Jackson laboratory (Bar Harbor, Maine). CD47KO-OVA-Tg B6 mice were generated by cross-breeding CD47KO B6 mice with OVA-Tg B6 mice. Simian diphtheria toxin receptor (DTR)-transgenic B6.FVB-Tg (Itagx-DTR/eGFP) 57Lan/J (CD11c-DTR) mice were purchased from the Jackson laboratory. All mice were bred in specific pathogen-free facilities. Protocols involving animals were approved by the Institutional Animal Care and Use Committees of the First Hospital of Jilin University and Columbia University, and all animal experiments were performed in accordance with the protocols.

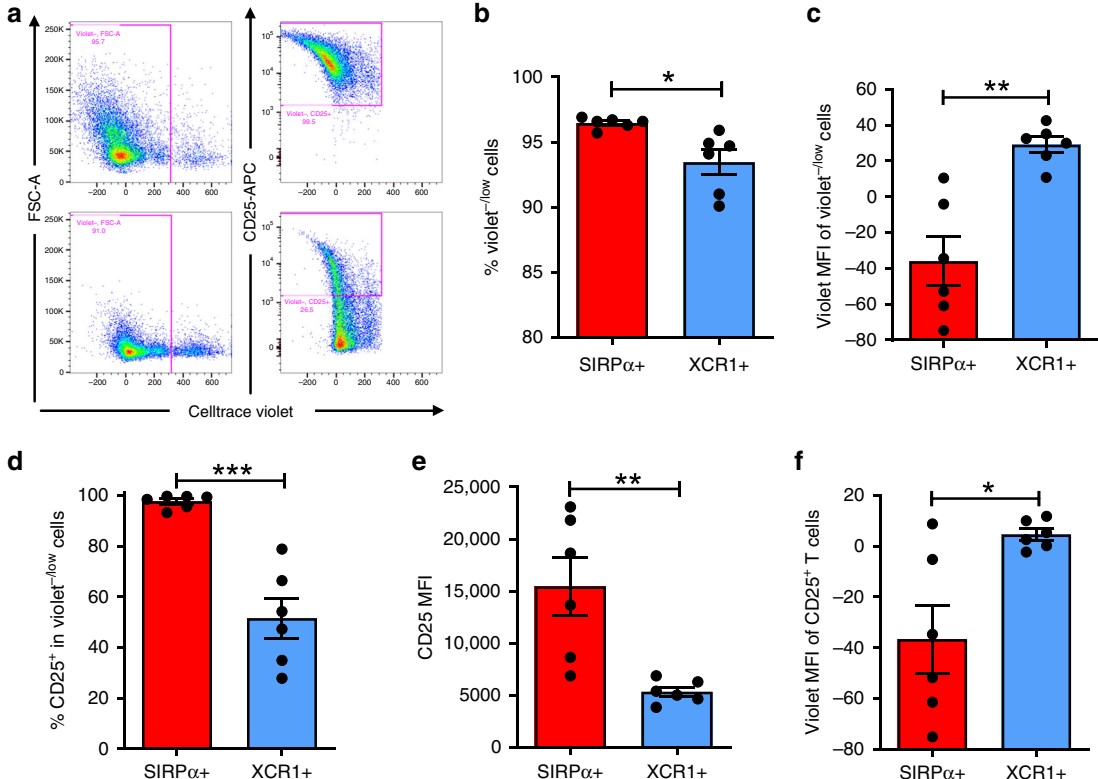

**Fig. 6 SIRPα⁺ DCs are superior to SIRPα⁻ DCs in stimulating T cell responses.** Flow cytometry-sorted splenic SIRPα⁺ (XCR1⁻CD11c⁺I-Ab⁺F4/80⁻CD64⁻) or XCR1⁺ (SIRPα⁻CD11c⁺I-Ab⁺F4/80⁻CD64⁻) DCs (Supplementary Fig. 5) were pulsed with 1nM OVA peptides (aa257-264), co-cultured with sorted OT-I mouse splenic CD3⁺CD8⁺ T cells (pre-labeled with Celltrace™ violet) for 3 days, then analyzed by flow cytometry for Violet fluorescence dilution in T cells (i.e., proliferation) and CD25 expression on Violet⁻/low T cells. **a** Representative flow cytometry profiles showing Violet fluorescence dilution in T cells (left) and CD25 expression on gated Violet⁻/low T cells (right) that were stimulated with SIRPα⁺ (top) or XCR1⁺ (bottom) DCs. **b** Percentages of Violet⁻/low T cells. **c** Violet MFI (mean fluorescence intensity) of Violet⁻/low T cells. **d** Percentages of CD25+ cells in Violet⁻/low T cells. **e** CD25 MFI of CD25⁺Violet⁻/low T cells. **f** Violet MFI of gated CD25⁺ T cells. Results are combined from two independent experiments (total n = 6 per group) and presented as mean ± SEM. *p < 0.05; **p < 0.01; ***p < 0.005 (unpaired t-test). Source data are provided as a Source Data file.

**Cell lines**. All cell lines were purchased from ATCC. B16F0 and B16F10 are C57BL/6 mouse-derived melanoma cell lines. EL4 is a C57BL/6 mouse-derived T lymphocyte lymphoma cell line. E.G7-OVA was derived from EL4 by transfection with the model antigen ovalbumin (OVA). These cell lines were tested negative for Mouse Essential Panel with PVM, provided by Charles River laboratory. All cells were grown in Dulbecco's Modified Eagle's Medium (Gibco) + Glutamine (Sigma-Aldrich), 10% FBS (Atlanta Biologicals) and 100 U/ml penicillin and streptomycin (Gibco).

**Flow cytometric analysis**. CD47 expression on tumor cells was determined by direct staining with AF647-conjugated rat anti-mouse CD47 antibody (clone MIAP301; rat IgG), or indirect staining with non-conjugated MIAP301 followed by goat anti-rat IgG antibody (clone Poly4054). For analysis of immune cell phenotypes, single cell suspensions were incubated with anti-mouse CD16/32 FcR blocking antibody (clone 93) and various combinations of the following fluorochrome-conjugated antibodies: anti-CD45 (clone 30-F11), CD11c (clone N418), CD11b (clone M1/70), SIRPα (CD172a; clone P84), CD3 (clone 17A2), CD4 (clone GK1.5), CD8 (clone 53-6.7), CD44 (clone IM7), CD62L (clone MEL-14), CD64 (clone X54-5/7.1), F4/80 (clone BM8), XCR1 (clone ZET), I-Ab (clone AF6-120.1), CD40 (clone 3/23), and CD86 (clone GL-1). Dead cells were identified by staining with propidium iodide or DAPI. All samples were collected on FACS Flow Cytometer (Fortessa, Becton Dickinson) and data were analyzed by Flowjo software (Tree Star). FACS sequential gating/sorting strategies are presented in Supplementary Fig. 9.

**CRISPR/Cas9-targeted cd47 gene inactivation in tumor cells**. CRISPR small guide RNA (sgRNA) targeting the first exon region of mouse CD47 was designed using the online tools (https://crispr.mit.edu) and was of the following sequence: sgCD47: TTGGCGGCGGCGCTGTTGCT. CD47⁻/⁻ B16F0 cell line was generated by transient transfection of CD47-targeting pSpCas9 (BB)-2A-GFP PX458 vector (a gift from Feng Zhang; Addgene plasmid #48138) encoding the above CD47-targeting sgRNA. The GFP⁺ cells were sorted by Influx cell sorter (BD Bioscience)

at 24 h post-transfection, cultured for 7 days, and CD47 expression was detected by staining with AF647-conjugated anti-CD47 antibody (MIAP301). The CD47⁻/⁻ B16F0 cell line was established by three rounds of cell sorting.

**Titration of anti-CD47 mAb for blocking CD47 on cell surface**. Non-conjugated rat anti-mouse CD47 mAb (MIAP301; IgG) was used to block cell surface CD47. Briefly, WT B16F0 tumor cells or OVA-Tg splenocytes were coated with two-fold serially diluted the blocking anti-CD47 antibody MIAP301 at 4 °C for 1 h, washed and stained with AF647-conjugated anti-CD47 antibody MIAP301 or AF647-conjugated goat anti-rat IgG antibody at 4° for 30 min, then the fluorescence intensity of AF647 was measured by flow cytometry. A lower fluorescence intensity indicates greater blocking effect.

**CD11c⁺ cell depletion in CD11c-DTR bone marrow chimeras**. Eight-week-old B6 mice were received 10.25 Gy total body irradiation (TBI) followed 8 h later by reconstitution with 1 × 10⁷ syngeneic BM cells from CD11c-DTR B6 mice. These BM chimeras were used for tumor challenge and vaccination 9 weeks later. In vivo CD11c⁺ cell depletion was performed by injection i.p. of diphtheria toxin (DT, 4 ng/gram body weight; Sigma-Aldrich) 12 h prior to and 24 h after B16F0 tumor cell vaccination, and the depletion efficacy was confirmed by flow cytometric analysis of CD11c⁺ cells in the spleen 1 day after second DT injection.

**Tumor inoculation and vaccination therapy**. For the B16F0 and B16F10 melanoma tumor models, tumor cell vaccination was performed in WT B6 mice by intrasplenic injection of 2.5 × 10⁶ CD47KO, WT or anti-CD47 mAb-coated WT melanoma cells, which were treated by 70 Gy irradiation or Mitomycin C (40 μg/ml for 2 h) immediately before injection. These mice were challenged by subcutaneous injection of WT melanoma (B16F0 or B16F10) cells in the right flank 1 day or 3 days before (1–2 × 10⁵ per mouse) or 7 days after (2 × 10⁵ per mouse) tumor cell vaccination to evaluate antitumor responses. The tumor dimensions were measured and the tumor volumes were calculated as $a \times b^2/2$ (a, length; b, width).

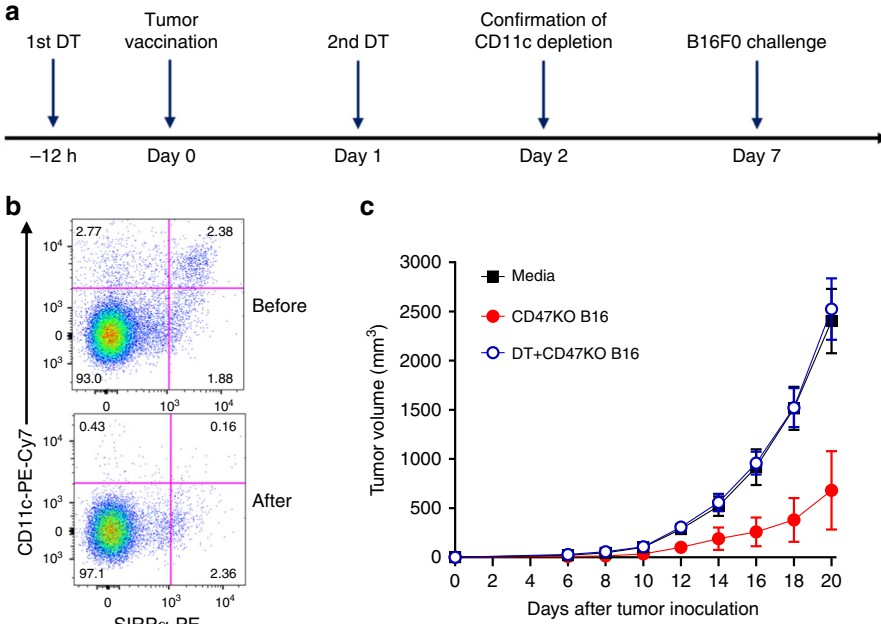

**Fig. 7 Role of CD11c+ DCs in antitumor effects of tumor cell vaccination. a** Schematic outline of the experimental design. **b** Flow cytometry profiles showing deletion of CD11c+ DCs in DT-injected mice (analysed 1 day after the last DT injection). **c** Tumor growth in CD11c-DTR→B6 BM chimeras that were vaccinated with media, or with irradiated CD47KO B16F0 with or without DT treatment ($n = 5$ per group), followed 7 days later by subcutaneous injection of $2 \times 10^5$ WT B16F0 melanoma cells. Tumor growth is shown as tumor volumes (mean ± SEM) over time. Source data are provided as a Source Data file.

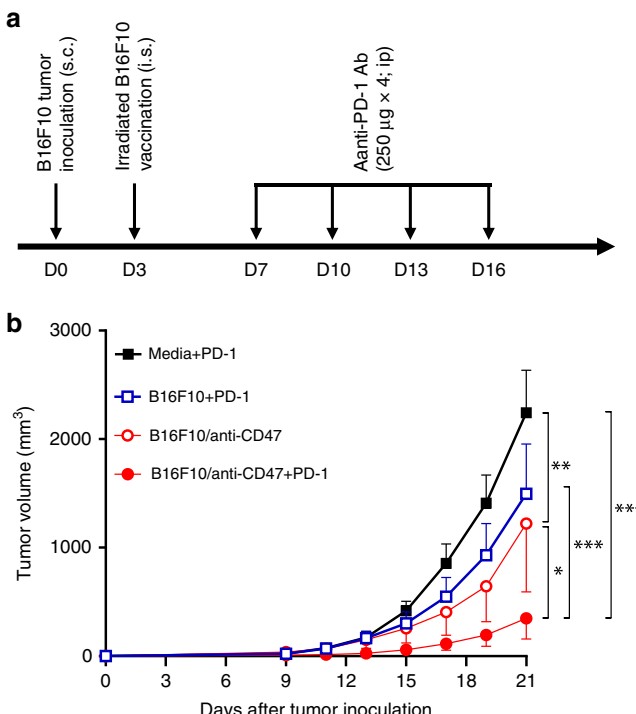

**Fig. 8 Synergistic antitumor effect of anti-CD47-coated tumor cells and anti-PD-1 Ab. a** Schematic outline of the experimental design. **b** Tumor growth in B16F10 tumor-bearing mice that received media plus anti-PD-1 ($n = 5$), vaccination with irradiated B16F10 plus anti-PD-1 ($n = 8$), with irradiated/anti-CD47 Ab-coated B16F10 ($n = 7$), or with irradiated/anti-CD47 Ab-coated B16F10 plus anti-PD-1 ($n = 8$). Tumor growth is shown as tumor volumes (mean ± SEM) over time. *$p < 0.05$; **$p < 0.01$; ***$p < 0.001$ (two-way ANOVA). Source data are provided as a Source Data file.

Where indicated, mice in some groups were also treated with 4 injections (250μg; i.p.) of anti-PD-1 antibodies (clone RMP1-14, Bio X cell). In some experiments, tumor-free mice were re-challenged by subcutaneous injection of $2 \times 10^5$ WT B16F0 cells to determine memory antitumor responses. For the EG7 lymphoma model, B6 mice were vaccinated by intrasplenic or intravenous injection of $1 \times 10^7$ OVA-expressing splenocytes from WT or CD47KO OVA-Tg mice. These mice were challenged with EG7 cells 1 day prior to ($1 \times 10^6$ per mouse; i.v.) or 7 days after ($2.5 \times 10^6$ per mouse; i.v.) vaccination with OVA-expressing splenocytes. In some experiments, long-term survivors were re-challenged by $2.5 \times 10^6$ EG7 (OVA+) or $2.5 \times 10^6$ EL4 (OVA−) cells to determine tumor antigen-specific memory responses.

**In vivo killing assay**. WT B6 mice were immunized by intrasplenic injection of $1 \times 10^7$ splenocytes from WT or CD47KO OVA-Tg B6 mice or PBS, followed 10 days by intravenous injection of a mixture (1:1) of CFSE-labeled OVA+ (CFSEhi) and OVA− (CSFElo) B6 mouse splenocytes. The ratio of OVA+ (CFSEhi) to OVA− (CSFElo) cells in the spleen was determined by flow cytometry 16 h after cell injection, and specific killing (%) of OVA+ cells was calculated as [percentage CFSEhi in the inoculum − percentage CFSEhi in the spleen)/(percentage CFSEhi in the inoculum)] × 100.

**Measurement of IFN-γ-secreting splenocytes by ELISPOT assay**. Spleen cells were prepared from WT B6 mice 7 days after intrasplenic injection of $2.5 \times 10^6$ 70 Gy-irradiated CD47KO or WT B16F0 cells or media, and measured for IFN-γ producing cells by ELISPOT assay according to the manufacturer's protocol (Mabtech). Briefly, 96-well MAIPS45 plate (Millipore) was pre-coated with 15 μg/ml anti-IFN-γ antibody (AN18) overnight at 4 °C, $2 \times 10^5$ splenocytes were co-cultured with 70 Gy-irradiated WT B16F0 cells at the ratio of 10:1 for 36 h then removed, 1 μg/ml detection anti-IFN-γ antibody (R4-6A2-biotin) was added, and plate was incubated for 2 h at room temperature. Streptavidin-ALP with 1:1000 dilution was added and the plate was incubated for 1 h at room temperature. Then substrate solution BCIP-NBT plus was added and develop until distinct spots emerge.

**Measurement of the capacity of DCs to stimulate T cells**. SIRPα+ (F4/80−CD64−CD11c+I-Ab+XCR1−) and SIRPα− (F4/80−CD64−CD11c+I-Ab+XCR1+) DCs were sorted from B6 mouse spleens by flow cytometry, and pulsed with 1 nM or 100 pM OVA peptides (aa257-264). CD3+CD8+ T cells were sorted from OT-I mouse spleens by flow cytometry and labeled with CelltraceTm violet (Thermo Fisher Scientific). The OVA-pulsed DCs ($1 \times 10^4$) were co-cultured with CelltraceTM violet-labeled OT-I CD8+ T cells ($1 \times 10^5$) for 3 days, and the cultured cells were analyzed

for proliferation and activation by measuring violet fluorescence dilution and CD25 expression, respectively, using flow cytometry.

**In vitro killing assay**. Spleen cells were prepared from WT B6 mice 7 days after intrasplenic injection of $2.5 \times 10^6$ 70 Gy-irradiated CD47KO or WT B16F0 cells or media, and measured for killing against B16F0 cells in vitro. Briefly, B16F0 cells were labeled with green fluorescent dye DiOC18(3) and incubated with the isolated splenic $CD3^+$ T cells. Five hours later, the cells were harvested and percentages of dead (i.e., $PI^+$) cells in tumor cells (i.e., $CD45^-DioC^+$ cells) were measured by flow cytometry.

**Statistical analysis**. Data were analyzed using GraphPad Prism (version 7; San Diego, CA) and presented as mean values ± SEM or mean values ± SD. Mouse survival data were presented as Kaplan-Meier survival curves and differences among groups were analyzed by the log-rank test. Tumor growing curves were analyzed by two-way ANOVA among groups. The level of significant differences in group means was assessed by student's $t$-test, and a $p$ value of ≤0.05 was considered significant in all analysis herein.

**Reporting summary**. Further information on research design is available in the Nature Research Reporting Summary linked to this article.

## Data availability

The authors declare that the data supporting the findings of this study are available within the article, supplementary information files and upon reasonable requests to the authors. The source data underlying Figs. 1b–f, 2a–e, 3b–e, 4b–e, 5b–h, 6b–f, 7c, and 8b and Supplementary Figs. 2a, 2b (left), 3c, 4a, and 8b are provided as a Source Data file.

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

## Acknowledgements

This work was supported by grants from the Ministry of Science and Technology of China (2015CB964400), National Institute of Health (AI 064569), and the Natural Science Foundation of China (91642208 and 81501383). Flow Cytometric analysis was performed in the CCTI Flow Cytometry Core funded in part through an NIH Shared Instrumentation Grant (1S10RR027050).

## Author contributions

Y.L., M.Z., and X.W. performed experiments; Y.L., M.Z., X.W., W.L., H.W. and Y.-G.Y. designed experiments and analyzed data; Y.-G.Y. conceived the research project and directed the research; Y.L., M.Z. and Y.-G.Y. wrote the paper; all authors edited and approved the paper.

## Competing interests

The authors declare no competing interests.

**Additional information**

