## [Peer Review File · Nature Communications]

Reviewers' comments:

Reviewer #1 (Remarks to the Author):

This study describes a novel approach to enhance the immunogenicity of whole tumor cell vaccines by blocking the negative effect CD47 expression on their recognition and uptake by the professional antigen processing network. An advantage of using whole tumor cell vaccines is that they present the full repertoire of tumor antigens, including the elusive neoantigens. From a translational standpoint the main disadvantage of whole tumor cell vaccines is the limited availability of autologous tumor cells for vaccine preparation; use of allogeneic and readily available tumor cells which will not present neoantigens to the patient' immune systems, has failed clinically.

This study does not make a strong case for their proposed approach.

1. The immunotherapy experiments in the B16F0 mode leave the impression that the therapeutic impact of CD47 blockade on the tumor vaccine is small. (a) The B16F10 subline is the most immunogenic subline in the series; most studies use the poorly immunogenic B16.F10 subline. (2) Pronounced effect was seen in a prophylactic setting and in the experiment performed in "therapeutic" setting vaccination started one day after tumor implantation (3) Protective memory was partial – strongest indication of poor immunogenicity of the vaccine.
2. Vaccination with OVA expressing splenocytes against E.G7-OVA tumor, a most immunogenic tumor, is not informative.
3. Vaccination with CD47 Ab coated tumor cells – the clinically relevant scenario- was marginally effective (in this otherwise immune susceptible model).
4. Intrasplenic vaccination is not readily translatable to human therapy. Is it necessary; is it so much superior to easier administration methods? This has not been addressed.

Reviewer #2 (Remarks to the Author):

In this manuscript, "Elicitation of potent antitumor immunity after intrasplenic vaccination with CD47-deficient tumor cells", Li et al reported that elimination of CD47 from tumor cells could significantly improve the effectiveness of whole tumor cell vaccination in mouse solid and hematopoietic tumor models. By utilizing CRISPR-Cas9 technique to conduct genetic deletion of CD47 or anti-CD47 mAb for blockade, intrasplenic vaccination with either tumor cells or tumor antigen-expressing splenocytes lacking CD47 or with CD47 blocking by antibodies displayed strong and durable antitumor immunity compared to their WT control. They also demonstrated that the induction of this antitumor response was highly dependent on SIRPα+CD11c+ DCs, since depleting CD11c+ DCs abrogated this anti-tumor immunity. Although their data demonstrated the effectiveness of improving whole tumor cell vaccination therapy with CD47 deficiency, there are still several problems to be addressed and the mechanism of how CD47-deficient tumor cells could induce a stronger response remained untold. Hence, more evidence should be provided to support their conclusion.

Functionally, although the vaccination of CD47-deficient cells could prevent the tumor growth post-vaccination and pre-inoculated tumor (one day inoculation), the significance of this finding would be greatly increased by examining whether the vaccination of CD47 deficiency cells could be used to induce anti-tumor effect against pre-existing tumor (ie, several days after inoculation of tumor cells to mimic establishing tumor).

Specific comments:

This manuscript showed that the anti-tumor response of the vaccination of CD47-deficient tumor cells against different tumor models is dependent on SIRPα+CD11c+ DCs. But the mechanism of how CD47 deficiency could regulate the expansion of SIRPα+CD11c+ DCs, and how this subsets of DCs could enhance anti-tumor response was not investigated. Therefore, revealing this mechanism would greatly strengthen this manuscript.

More mechanistic exploration about the "antigen-specific" induction of antitumor response induced by this vaccination need to be addressed to complete this story. For instance, can those CD11c+SIRPα+ DCs endocytose more CD47KO B16F0 cells than WT B16F0 cells in vitro? If they can, will SIRPα-deficient DCs lose their capacity of promoting this antitumor immunity? Answering these questions could demonstrate the mechanism of the strong antitumor response induced by CD47-deficient tumor cell vaccination.

Authors found a rapid expansion in SIRPα+ cells and analyzed the phenotypes of SIRPα+ cell populations to focus on the CD11c+ DCs. They also used DC depletion to indicate their important role in this antitumor immune response. However, the provided data were not sufficient to draw the conclusion of "Rapid expansion of CD11c+SIRPα+ DCs is essential for ...". With DC depletion experiment, it would be better to provide more evidence to prove the correlation between DC expansion and antitumor immunity. Although the CD47- SIRPα axis may be crucial for DC function, but CD47- SIRPα axis also exists in macrophages. From the Figure S4B, the CD11b+ CD11c- cells had the most significant changes upon CD47KO cells vaccination. In addition, previous studies showed that macrophages are responsible for the CD47 antibody-associated anti-tumor effect. Therefore, authors need to examine the changes of other immune cells such as macrophages upon the vaccination of CD47KO cells.

In Figure 3, authors only showed the preventive effect of CD47 antibody-coated tumor cells. However, it would be more important to test the therapeutic effect of CD47 antibody-coated tumor cells as cancer vaccine after tumors established.

In Fig. 4, using only CD11c and CD11b were not precise enough to identify DCs, there were macrophages in these subsets, and SIRPα+ macrophages had also been reported to endocytose CD47- cells. Therefore, more macrophage-specific or DC-specific molecules should be used to distinguish between them.

In Fig. S1, except for IF staining, authors should use other methods such as FACS to evaluate the tumor cell distribution.

In the Discussion, authors stated "In line with that study, here we found that vaccination with CD47KO tumor cells or TA-expressing splenocytes induced a rapid CD11c+ DC activation and vigorous T cell-mediated antitumor responses". However, authors only showed the expansion of CD11c+ DCs, but not the activation markers increase such as CD40, CD86, CD80 upon CD47KO tumor cells challenge. Functionally, authors only showed an increased IFN-γ secretion of T cells and effector/memory T cells upon CD47KO tumor cell vaccination. It would be more convincing to isolate T cells from mice vaccinated with CD47KO tumor cells to perform an in vitro killing experiment for targeted cells.

We are grateful to the reviewers for their helpful and constructive comments. We have performed several new experiments and revised the manuscript in response to these comments and suggestions. The new experimental data are presented in Figs 3d, 3e, 5, 6, and 8 and Supplementary Figs S2a, S5, S6, and S8 in the revised manuscript. The following is a point-by-point response to the reviewers' comments (Changes in the revised manuscript are highlighted by red-colored font):

Reviewers' comments:

Reviewer #1 (Remarks to the Author):

This study describes a novel approach to enhance the immunogenicity of whole tumor cell vaccines by blocking the negative effect CD47 expression on their recognition and uptake by the professional antigen processing network. An advantage of using whole tumor cell vaccines is that they present the full repertoire of tumor antigens, including the elusive neoantigens. From a translational standpoint the main disadvantage of whole tumor cell vaccines is the limited availability of autologous tumor cells for vaccine preparation; use of allogeneic and readily available tumor cells which will not present neoantigens to the patient's immune systems, has failed clinically.

This study does not make a strong case for their proposed approach.

Response: We thank the reviewer for recognizing the novelty of our approach. We believe that our revised manuscript, in which we have addressed the most key comments raised by the reviewer, has now been significantly improved.

1. The immunotherapy experiments in the B16F0 mode leave the impression that the therapeutic impact of CD47 blockade on the tumor vaccine is small. (a) The B16F10 subline is the most immunogenic subline in the series; most studies use the poorly immunogenic B16.F10 subline. (2) Pronounced effect was seen in a prophylactic setting and in the experiment performed in "therapeutic" setting vaccination started one day after tumor implantation (3) Protective memory was partial – strongest indication of poor immunogenicity of the vaccine.

Response: We thank the reviewer for pointing out these issues. For Comment (a), as the

reviewer suggested, we have performed new experiments to evaluate the efficacy of CD47-deficient tumor vaccination in the B16F10 model. Our results from multiple experiments confirmed that our strategy can also induce significant antitumor responses in B16F10 tumor-bearing mice (as shown in Fig 3D, Fig 3E and Fig 8 in the revised manuscript).

To address the reviewer's comment (2), we performed new experiments to assess the therapeutic antitumor efficacy, in which vaccination was given 3 days after tumor cell implantation. We confirmed that vaccination with anti-CD47 ab-coated, but not non-coated, WT tumor cells can also induce significant antitumor effects in this therapeutic model (Fig 8 in the revised manuscript).

For comment (3), as the reviewer noted that this approach could not induce a memory response that is sufficient to achieve 100% complete protection against secondary tumor challenge. In the B16F0 model, memory responses, as shown by protection against second tumor challenge, were detected in all animals vaccinated with CD47KO tumor cells, with approximately 30% (2/7) mice showing complete protection (Figure 1F). Strong memory responses were detected in the EG7 model, in which 7/10 (70%) of tumor-free mice vaccinated with CD47KO cells survived tumor rechallenge for additional 100 days until sacrifice (Figure 2D). These results provided convincing evidence demonstrating that memory antitumor responses were induced by a single vaccination with CD47-deficient tumor cells, despite that complete protection against secondary tumor challenge was not achieved in all animals. As discussed throughout the manuscript, whole tumor cell vaccination is a very simple approach, but tumor cells are poorly immunogenic and are shown to secrete soluble factors capable of suppressing DC differentiation and maturation^{1,2}. Thus, it is felt that deletion or antibody-coating of CD47 offers a novel approach to increasing the efficacy of whole tumor vaccination, which would have a great impact on cancer immunotherapy, in particularly if its efficacy can be further enhanced by combination therapy. In the revised manuscript, we have now included new experimental data demonstrating that this approach can act synergistically with other forms of immunotherapy (e.g., anti-PD-1 antibodies) to further enhance the antitumor efficacy in mice with established tumor (Figure 8 in the revised manuscript).

2. Vaccination with OVA expressing splenocytes against E.G7-OVA tumor, a most immunogenic tumor, is not informative.

Response: We agree with the reviewer that E.G7 is an immunogenic model (due to expression of

an artificial model antigen, OVA). However, this model has been widely used to study cancer immunology and immunotherapy, and in many cases, the results obtained from this tumor model are representative of what observed in models using the parental mouse tumor cell lines. Moreover, when used in combination with its parental cell line EL4, this model makes it possible to precisely assess antigen (OVA)-specific T cell responses (see Figure 2D and 2E in the revised manuscript). We agree that, because of the relatively high immunogenicity, data from this model alone are insufficient in making a firm conclusion and therefore, multiple tumor models (including B16F0 and B16F10 in addition to E.G7) were used in our studies.

3. Vaccination with CD47 Ab coated tumor cells – the clinically relevant scenario- was marginally effective (in this otherwise immune susceptible model).

Response: In the E.G7 model, while lower than mice vaccinated with CD47KO cells, vaccination with anti-CD47 ab-coated cells achieved long-term tumor-free survival in approximately 60% mice challenged with a lethal dose of tumor cells. Similarly, vaccination with anti-CD47 Ab-coated tumor cells also induced a significant antitumor response in B16F0 model, although the efficacy was relatively lower than CD47KO tumor cell vaccination. Furthermore, the antitumor effect of anti-CD47 Ab-coated tumor cell vaccine was further confirmed in our new experiments using B16F10. As shown in Figure 3D and Figure 8 in the revised manuscript, vaccination with anti-CD47 ab-coated B16F10 tumor cells induced significant antitumor responses in both prophylactic (Figure 3D) and therapeutic (Figure 8) conditions (please refer to our responses to the reviewer’s comment #1 above). As discussed in the revised manuscript, the lower protection by anti-CD47-coated WT tumor cells than CD47KO tumor cells may possibly be due to quick antibody dissociation from CD47 on tumor cells after intrasplenic injection into mice. In support of this possibility, we found that anti-CD47-coated tumor cell vaccination appeared to be more effective when given subcutaneously than when given intrasplenically (Figure 3D and 3E). It is perceivable that antibody release from coated tumor cells is expected to be more efficient (due to rapidly entering circulation) following intrasplenic injection than subcutaneously local injection. These results also demonstrate that subcutaneous vaccination with tumor cells lacking surface CD47 can also induce significant antitumor responses.

4. Intrasplenic vaccination is not readily translatable to human therapy. Is it necessary; is it so much superior to easier administration methods? This has not been addressed.

Response: We thank the reviewer for raising this important comment. As the reviewer suggested, we have performed new experiments to investigate the potential of subcutaneous vaccination to induce antitumor responses. As shown in Figure 3E in the revised manuscript, subcutaneous immunization with anti-CD47 ab-coated B16F10 cells also induced significant antitumor responses. In the EG.7 model, significant protection was also observed in mice receiving intravenous vaccination (though the magnitude was not as efficient as IS vaccination). Taken together, these results indicate that the antitumor response of CD47-deficient tumor cell vaccine can be achieved by different routes of vaccination, including subcutaneous and intravenous injections (does not have to be given intrasplenically).

Reviewer #2 (Remarks to the Author):

In this manuscript, “Elicitation of potent antitumor immunity after intrasplenic vaccination with CD47-deficient tumor cells”, Li et al reported that elimination of CD47 from tumor cells could significantly improve the effectiveness of whole tumor cell vaccination in mouse solid and hematopoietic tumor models. By utilizing CRISPR-Cas9 technique to conduct genetic deletion of CD47 or anti-CD47 mAb for blockade, intrasplenic vaccination with either tumor cells or tumor antigen-expressing splenocytes lacking CD47 or with CD47 blocking by antibodies displayed strong and durable antitumor immunity compared to their WT control. They also demonstrated that the induction of this antitumor response was highly dependent on SIRP α +CD11c+ DCs, since depleting CD11c+ DCs abrogated this anti-tumor immunity. Although their data demonstrated the effectiveness of improving whole tumor cell vaccination therapy with CD47 deficiency, there are still several problems to be addressed and the mechanism of how CD47-deficient tumor cells could induce a stronger response remained untold. Hence, more evidence should be provided to support their conclusion.

Functionally, although the vaccination of CD47-deficient cells could prevent the tumor growth post-vaccination and pre-inoculated tumor (one day inoculation), the significance of this finding would be greatly increased by examining whether the vaccination of CD47 deficiency cells could be used to induce anti-tumor effect against pre-existing tumor (ie, several days after inoculation of tumor cells to mimic establishing tumor).

Response: We thank the reviewer for recognizing the significance of our findings. As detailed

below, we have performed more mechanistic studies in response to the reviewer's comments. We also performed new experiments to confirm the antitumor responses against pre-inoculated tumor, in which the tumor was established by implantation of tumor cells 3 days before vaccination (results are presented in Figure 8 in the revised manuscript; please also refer to our response to the reviewer's comment #4 below).

Specific comments:

1. This manuscript showed that the anti-tumor response of the vaccination of CD47-deficient tumor cells against different tumor models is dependent on SIRP α +CD11c+ DCs. But the mechanism of how CD47 deficiency could regulate the expansion of SIRP α +CD11c+ DCs, and how this subsets of DCs could enhance anti-tumor response was not investigated. Therefore, revealing this mechanism would greatly strengthen this manuscript.

Response: We thank the reviewer for raising this important issue and have performed new experiments to address the reviewer's concern. In the previous manuscript, we reported that a significant expansion of SIRP α +CD11c+ DCs was detected in mice vaccinated with CD47-deficient tumor cells compared to those vaccinated with WT tumor cells (Figure 4 in the revised manuscript). To further rule out the possibility of contamination by monocytes/macrophages, we recently repeated this experiment, in which expansion and activation of SIRP α +CD11c+ DCs were examined after gating out CD64+ and F4/80+ monocytes/macrophages. As shown in Figure 5 in the revised manuscript, significantly increased expansion and activation (as shown by increased expression of I-Ab, CD40 and CD86) of SIRP α +CD11c+ DCs (macrophages were gated out from DC analysis by CD64 and F4/80 expression) were detected in mice vaccinated with CD47KO tumor cells compared to those received WT tumor cell vaccination. In order to understand how this DC subset could enhance anti-tumor responses, we further compared the ability to prime and activate T cells of SIRP α + vs. SIRP α - DCs. It has been reported that splenic DCs consist of mainly two conventional DC subsets, i.e., SIRP α +XCR1- and SIRP α -XCR1+ DCs³. Thus, we sorted SIRP α + (F4/80⁻CD64⁻CD11c⁺I-Ab⁺XCR1⁻) and SIRP α - (F4/80⁻CD64⁻CD11c⁺I-Ab⁺XCR1⁺) DCs (Figure S5), pulsed the DCs with OVA peptides (aa257-264), and analyzed for their ability to activate CD3+CD8+ T cells isolated from OT-I mice. As shown in Figure 6 and S6 in the revised manuscript, SIRP α + DCs were significantly more effective than SIRP α - DCs in inducing CD8 T cell proliferation and CD25 upregulation. These new data provide a further mechanistic link

between increased SIRP α +CD11c+ DC expansion/activation and enhanced antitumor T cell responses.

2. More mechanistic exploration about the “antigen-specific” induction of antitumor response induced by this vaccination need to be addressed to complete this story. For instance, can those CD11c+SIRP α + DCs endocytose more CD47KO B16F0 cells than WT B16F0 cells in vitro? If they can, will SIRP α -deficient DCs lose their capacity of promoting this antitumor immunity? Answering these questions could demonstrate the mechanism of the strong antitumor response induced by CD47-deficient tumor cell vaccination.

Response: We thank the reviewer for raising this important question. In the revised manuscript, we have included our new data demonstrating that CD11c+SIRP α + DCs were more effective in endocytosing CD47KO B16F0 than WT B16F0 cells (see Figure S8 in the revised manuscript). This study suggests that the lack of CD47-SIRP α signaling results in augmentation of antitumor responses by enhancing DC endocytosis of target cells and T cell priming. Since SIRP α -deficient DCs are also defective in SIRP α signaling, these DCs should have an enhanced capacity of inducing T cell responses ⁴.

3. Authors found a rapid expansion in SIRP α + cells and analyzed the phenotypes of SIRP α + cell populations to focus on the CD11c+ DCs. They also used DC depletion to indicate their important role in this antitumor immune response. However, the provided data were not sufficient to draw the conclusion of “Rapid expansion of CD11c+SIRP α + DCs is essential for ...”. With DC depletion experiment, it would be better to provide more evidence to prove the correlation between DC expansion and antitumor immunity. Although the CD47- SIRP α axis may be crucial for DC function, but CD47- SIRP α axis also exists in macrophages. From the Figure S4B, the CD11b+ CD11c- cells had the most significant changes upon CD47KO cells vaccination. In addition, previous studies showed that macrophages are responsible for the CD47 antibody-associated anti-tumor effect. Therefore, authors need to examine the changes of other immune cells such as macrophages upon the vaccination of CD47KO cells.

Response: As the reviewer pointed out that significant expansion of CD11b+CD11c- cells (monocytes/macrophages) was also observed in mice injected with CD47KO splenocytes compared to those receiving WT splenocytes (Figure S4B). This is in line with our previous studies, in which expansion of SIRP α + DCs and macrophages were observed in mice infused

with CD47KO splenocytes compared to those receiving WT splenocytes in both syngeneic and allogeneic settings^{5,6,7}. However, vaccination with CD47KO tumor cells only induced expansion and activation of SIRP α +CD11c+ DCs (Figures 4C, 4E, 5B, and 5C in the revised manuscript), but not SIRP α +CD11c- (Figure 4D) or CD64+F4/80+ (Figure 5D) monocytes/macrophages, supporting our conclusion that SIRP α +CD11c+ DCs are essential in the induction of antitumor T cell responses by CD47KO tumor cell vaccination. Furthermore, as discussed in our response to the reviewer's comment #1, our new data provide a direct link between increased SIRP α + DC activation and enhanced T cell responses (Figure 6 and S6 in the revised manuscript). Together, these observations, plus the facts that the improved antitumor responses and enhanced SIRP α +CD11c+ DC expansion/activation in CD47KO tumor-vaccinated mice, and that CD47KO tumor vaccination failed to induce antitumor responses in CD11c+ DC-depleted mice, suggest that the effect of CD47KO tumor vaccination is predominantly dependent on SIRP α +CD11c+ DCs.

4. In Figure 3, authors only showed the preventive effect of CD47 antibody-coated tumor cells. However, it would be more important to test the therapeutic effect of CD47 antibody-coated tumor cells as cancer vaccine after tumors established.

Response: As the reviewer suggested, we recently evaluated the therapeutic effect of CD47 antibody-coated tumor cell vaccination, in which mice were vaccinated 3 days after subcutaneous inoculation of B16F10 tumor cells. As shown in Figure 8 in the revised manuscript, vaccination with CD47-coated tumor cells induced a significant antitumor response, which was further synergistically enhanced by treatment with anti-PD-1 antibodies, in this therapeutic setting.

5. In Fig. 4, using only CD11c and CD11b were not precise enough to identify DCs, there were macrophages in these subsets, and SIRP α + macrophages had also been reported to endocytose CD47- cells. Therefore, more macrophage-specific or DC-specific molecules should be used to distinguish between them.

Response: We agree with the reviewer and have performed new analyses, in which macrophages were discriminated by CD64 and F4/80 expression as previously reported³. Again, we found that CD47KO tumor vaccination induced a significant expansion and activation of SIRP α +CD11c+CD64-F4/80- DCs, but not CD64+F4/80+ macrophages (see Figure 5 in the

revised manuscript).

6. *In Fig. S1, except for IF staining, authors should use other methods such as FACS to evaluate the tumor cell distribution.*

Response: The presence of CFSE-labelled tumor cells in the spleen and liver was examined by both fluorescence microscopy and FACS, but the latter was less informative than the former analysis. This is mainly because the tumor cells were irradiated and therefore, the percentages of tumor cells were very low by FACS analysis after PI gating (i.e., gating out the dead cells). Thus, we only presented the fluorescence microscopy data in the manuscript.

7. *In the Discussion, authors stated “In line with that study, here we found that vaccination with CD47KO tumor cells or TA-expressing splenocytes induced a rapid CD11c+ DC activation and vigorous T cell-mediated antitumor responses”. However, authors only showed the expansion of CD11c+ DCs, but not the activation markers increase such as CD40, CD86, CD80 upon CD47KO tumor cells challenge. Functionally, authors only showed an increased IFN- γ secretion of T cells and effector/memory T cells upon CD47KO tumor cell vaccination. It would be more convincing to isolate T cells from mice vaccinated with CD47KO tumor cells to perform an in vitro killing experiment for targeted cells.*

Response: We thank the reviewer for pointing out these important issues, and performed new experiments as the reviewer suggested. In the revised manuscript, we have now included our new data showing that CD47KO vaccination induced significant expansion and activation (as shown by upregulation of I-Ab, CD40 and CD86) of SIRP α + DCs (Figure 5 in the revised manuscript). We also showed that these SIRP α + DCs were more efficient than SIRP α - DCs in stimulating antigen-specific T cell proliferation and activation (Figure 6 and Figure S6 in the revised manuscript).

For T cell function, we previously performed in vivo killing assay to measuring specific killing of OVA+ cells in mice vaccinated with OVA-expressing WT or CD47KO cells (Figure S3). As the reviewer suggested, we have now measured the killing of melanoma cells by T cells isolated from mice vaccinated with irradiated tumor cells. As shown in Figure S2a in the revised manuscript, T cells from mice vaccinated with CD47KO tumor cells were significantly more effective than those of WT tumor cell-vaccinated mice in killing of melanoma cells.

Note: As discussed above and in the manuscript, significant antitumor responses were not only achieved by intrasplenic vaccination, but also by other administration methods (e.g., subcutaneous injection); therefore, we have removed the word “intrasplenic” from the title of the manuscript.

References

1. Copier J, Dalgleish A. Whole-cell vaccines: A failure or a success waiting to happen? *Current opinion in molecular therapeutics* **12**, 14-20 (2010).
2. Chiang CL, Coukos G, Kandalaft LE. Whole Tumor Antigen Vaccines: Where Are We? *Vaccines (Basel)* **3**, 344-372 (2015).
3. Guilliams M, *et al.* Unsupervised High-Dimensional Analysis Aligns Dendritic Cells across Tissues and Species. *Immunity* **45**, 669-684 (2016).
4. Bian Z, *et al.* Cd47-Sirp α interaction and IL-10 constrain inflammation-induced macrophage phagocytosis of healthy self-cells. *Proceedings of the National Academy of Sciences* **113**, E5434-E5443 (2016).
5. Wang H, Wu X, Wang Y, Oldenborg PA, Yang YG. CD47 is required for suppression of allograft rejection by donor-specific transfusion. *Journal of immunology (Baltimore, Md : 1950)* **184**, 3401-3407 (2010).
6. Wang Y, Wang H, Bronson R, Fu Y, Yang YG. Rapid dendritic cell activation and resistance to allotolerance induction in anti-CD154-treated mice receiving CD47-deficient donor-specific transfusion. *Cell transplantation* **23**, 355-363 (2014).
7. Wang H, Madariaga ML, Wang S, Van Rooijen N, Oldenborg PA, Yang YG. Lack of CD47 on nonhematopoietic cells induces split macrophage tolerance to CD47null cells. *Proceedings of the National Academy of Sciences of the United States of America* **104**, 13744-13749 (2007).

REVIEWERS' COMMENTS:

Reviewer #1 (Remarks to the Author):

In the revised manuscript the authors went some way of addressing a main concern of this reviewer, namely that the therapeutic impact and clinical applicability of the proposed approach has been limited. In the revised manuscript they used also the more stringent B16.F10 murine tumor model (Fig. 3D) and have shown that subcutaneous, as opposed to intrasplenic, vaccination is also effective (Fig. 3E). Rest of the experiments remain as described in the original study and make use of highly immunogenic models (E.G7-OVA and B16.F0) that are minimally informative.

Reviewer #2 (Remarks to the Author):

Authors have addressed my concerns. I do not have more questions.

Point-by-point response to reviewers' comments

Reviewer #1 (Remarks to the Author):

In the revised manuscript the authors went some way of addressing a main concern of this reviewer, namely that the therapeutic impact and clinical applicability of the proposed approach has been limited. In the revised manuscript they used also the more stringent B16.F10 murine tumor model (Fig. 3D) and have shown that subcutaneous, as opposed to intrasplenic, vaccination is also effective (Fig. 3E). Rest of the experiments remain as described in the original study and make use of highly immunogenic models (E.G7-OVA and B16.F0) that are minimally informative.

Response: We thank this reviewer for the positive response to the revisions made to our previous manuscript.

Reviewer #2 (Remarks to the Author):

Authors have addressed my concerns. I do not have more questions.

Response: We thank the reviewer for the positive response to our revised manuscript.